# One-hour coherent optical storage in an atomic frequency comb memory

Yu Ma[1,2], You-Zhi Ma[1,2], Zong-Quan Zhou [1,2✉], Chuan-Feng Li [1,2✉] & Guang-Can Guo[1,2]

Photon loss in optical fibers prevents long-distance distribution of quantum information on the ground. Quantum repeater is proposed to overcome this problem, but the communication distance is still limited so far because of the system complexity of the quantum repeater scheme. Alternative solutions include transportable quantum memory and quantum-memory-equipped satellites, where long-lived optical quantum memories are the key components to realize global quantum communication. However, the longest storage time of the optical memories demonstrated so far is approximately 1 minute. Here, by employing a zero-first-order-Zeeman magnetic field and dynamical decoupling to protect the spin coherence in a solid, we demonstrate coherent storage of light in an atomic frequency comb memory over 1 hour, leading to a promising future for large-scale quantum communication based on long-lived solid-state quantum memories.

[1] CAS Key Laboratory of Quantum Information, University of Science and Technology of China, Hefei, PR China. [2] CAS Center For Excellence in Quantum Information and Quantum Physics, University of Science and Technology of China, Hefei, PR China. ✉email: zq_zhou@ustc.edu.cn; cfli@ustc.edu.cn

Quantum repeater[1] is proposed to establish the long-distance entanglement in the presence of channel loss. However, the construction of a practical quantum repeater is of great challenge due to its demanding requirements and system complexity[2-6]. Space-borne quantum communication[7,8] and transportable quantum memory[9] can avoid using optical fibers, but optical quantum memories with a lifetime on the order of hours would be essential for extending the communication distance to global scale.

With the help of a zero-first-order-Zeeman (ZEFOZ) magnetic field and dynamical decoupling (DD), Zhong et al. have reported a 6 h spin coherence lifetime[9] in europium-doped yttrium orthosilicate ($Eu^{3+}$:$Y_2SiO_5$). This long-lived spin coherence is a key resource for constructing quantum memories for global quantum communication. Although hours of spin coherence time have been demonstrated, long-lived optical storage remains a challenge because of the complicated and unknown energy structures in ZEFOZ fields[9-11] and a reduced effective absorption in magnetic fields[12] because only one subsite can be used for the long-term storage[11]. To date, the longest optical storage time is ~1 min realized in $^{87}$Rb atoms[13] and a $Pr^{3+}$:$Y_2SiO_5$ crystal using the electromagnetically induced transparency protocol[10]. For single-photon-level storage, the longest storage time is ~1 s realized in $^{87}$Rb atoms[14].

To take advantages of the long-lived spin coherence, a spin-wave based optical storage protocol should be employed. So far, atomic frequency comb (AFC)[15,16] has been the only successful protocol for spin-wave storage of photonic qubits[17-19] in the rare-earth-ion doped crystals. In $Eu^{3+}$:$Y_2SiO_5$, spin-wave storage of polarization qubit[20] and the generation and multimode storage of single photons as spin waves[21] have been demonstrated based on AFC protocol. A recent work extends the AFC memory time to 0.5 s in $Eu^{3+}$:$Y_2SiO_5$[22].

Here we demonstrate a coherent optical memory with a storage time of 1 h using a $^{151}Eu^{3+}$:$Y_2SiO_5$ crystal, based on the spin-wave AFC protocol[16] in a ZEFOZ field, namely ZEFOZ-AFC method. DD is used to protect the spin coherence and extend the storage time. The coherent nature of this device is verified by implementing a time-bin-like interference experiment after 1 h storage with a fidelity of 96.4%, which shows the feasibility of qubit storage. Our work improves the AFC memory time by ~6000 times[22].

## Results

### Characterization of the energy level structure.
In order to implement optical storage in a ZEFOZ field, the knowledge of the energy level structures in both the ground and excited state is a prerequisite. Previous works[11,23,24] have determined the spin Hamiltonians for the ground-state $^7F_0$ and excited-state $^5D_0$, which can be used to predict the level structures in any given magnetic field. However, the theoretical predictions may have an error comparable to the storage bandwidth and lead to a wrong choice of pumping strategies. In order to precisely determine the structures, we first find the ZEFOZ transition $|3\rangle_g \leftrightarrow |4\rangle_g$ by adjusting the strength of the field and orientation of the sample. We then use the continuous-wave Raman heterodyne detection (RHD)[25] to probe the ground-state resonances in the ZEFOZ field. However, continuous-wave RHD fails to detect the excited-state resonances due to the short-lived population in the optically excited states and the weaker interaction strength with the radio-frequency (RF) field. Here we employ the pulsed RHD to probe the excited-state resonances (see Supplementary Note 1). The experimentally determined energy level structure is presented in Fig. 1a.

**Experimental setup.** In order to increase the effective absorption, we use an isotopically enriched sample of $^{151}Eu^{3+}$:$Y_2SiO_5$ crystal (Scientific Materials), with a doping level of 1000 ppm and an isotopic enrichment of 99.3%. The sample has an optical depth $\alpha L$ of 2.6 for $Eu^{3+}$ at site 1. In order to quickly find the correct direction of the sample in the ZEFOZ field, the optical surface of the sample is cut perpendicularly to the direction of the known ZEFOZ magnetic field with a cylindrical shape, such that the field of the superconductor magnet along the vertical direction (Fig. 1b) is approximately aligned with the ZEFOZ field (a photo is provided in Supplementary Fig. 2b). The field is 1.280 T in the direction of $[-0.535, -0.634, 0.558]$ in the $[D_1, D_2, b]$ frame of the crystal as indicated in previous works[9,11]. The sample has a diameter of 4.5 mm and a thickness of 6 mm, and is mounted on two goniometers. The goniometers allow tilting in two dimensions, with a tilting range within 6.6° and a positioning resolution of 0.002°. A pair of 4-turn Helmholtz coils with a diameter of 6 mm is placed at the two sides of the sample. The coils are driven by an arbitrary waveform generator with the output amplified by an RF amplifier with an output power of 300 W, to implement the DD sequences. The experiments are conducted after the sample is cooled to 1.7 K.

The laser is locked to an ultra-stable plano-concave cavity at 580.039 nm, with a linewidth well below 10 kHz. The pump and probe beam are independently generated by acoustic-optic modulators in the double-pass configuration. Then the pump and probe beam are overlapped with each other inside the crystal with a diameter of 200 and 50 μm respectively, and an angle of ~30 mrad with each other. The beams double pass the crystal, and the transmitted probe beam is collected by a single-mode fiber. The probe beam is then combined with a reference laser at a locked frequency offset of 43.66 MHz for optical heterodyne detection. The beat signal is captured by a photodetector and demodulated to give the amplitude of the signals.

**Spin-wave AFC and dynamical decoupling.** The experimental sequence is shown in Fig. 2a. The experiment begins with a preparation procedure based on spectral hole burning[26] (see Supplementary Note 2). The first step is the so-called "class-cleaning", which can isolate a single class of ions in the inhomogeneously broadened sample. Six chirped pulses which are resonant with $|1\rangle_g \leftrightarrow |1\rangle_e$, $|2\rangle_g \leftrightarrow |2\rangle_e$, $|3\rangle_g \leftrightarrow |3\rangle_e$, $|4\rangle_g \leftrightarrow |3\rangle_e$, $|5\rangle_g \leftrightarrow |3\rangle_e$, and $|6\rangle_g \leftrightarrow |5\rangle_e$ transitions are applied to choose the target ions as shown in Fig. 1a. Other classes of ions will relax to other nonresonant ground-state levels and no longer interact with the six pulses. The chirp bandwidth of these pulses is 1 MHz. The second step is the so-called "spin polarization", which initializes all the population of the chosen ions into the state $|3\rangle_g$ by using the five pulses mentioned above except for $|3\rangle_g \leftrightarrow |3\rangle_e$. After these two steps, a Λ system has been well isolated, which comprises the level $|3\rangle_g$, $|4\rangle_g$, and $|3\rangle_e$. The third step is the "AFC preparation". An AFC is prepared in the $|3\rangle_g$ level by frequency-selected hole burning while keeping the $|4\rangle_g$ level empty. The AFC has a bandwidth of 1.0 MHz and a periodicity of $\Delta = 100$ kHz, which leads to an echo emission at $t = 1/\Delta = 10$ μs after input. After the AFC preparation, a 2 μs (full width at half maximum) probe pulse of Gaussian profile with a power of 150 μW resonant with $|3\rangle_g \leftrightarrow |3\rangle_e$ is absorbed by the sample and creates a collective excitation. A control pulse with a width of 4 μs and a power of 360 mW resonant with $|4\rangle_g \leftrightarrow |3\rangle_e$ immediately transfers the optical coherence into a spin-wave excitation. The control pulse has a complex hyperbolic secant profile[27] to achieve efficient control over the bandwidth of the AFC. Then a DD sequence is applied to protect the spin coherence. After that, a

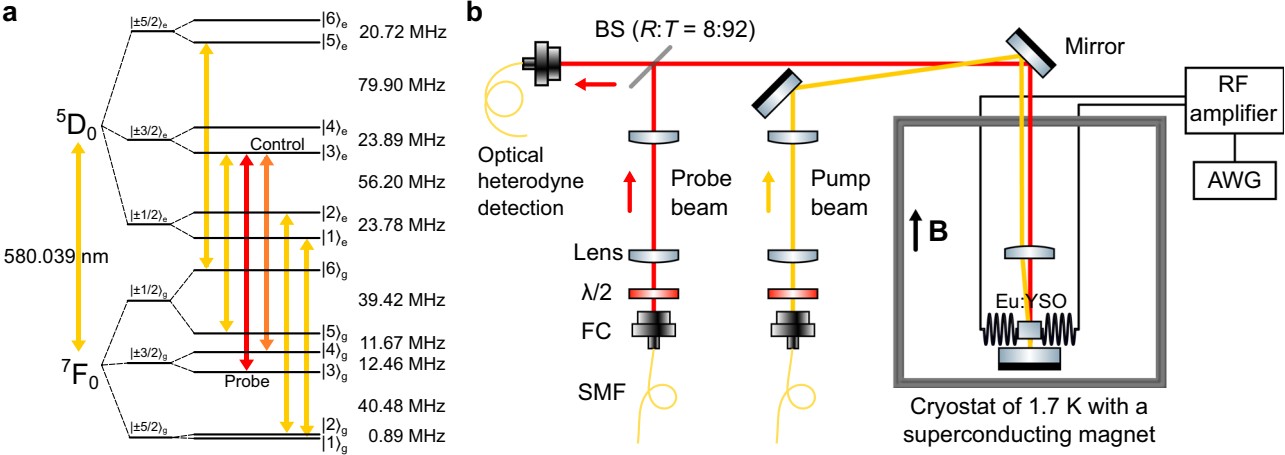

**Fig. 1 Energy level diagram and experimental setup. a** The laser at 580.039 nm is resonant with electronic state $^5D_0$ and $^7F_0$ of $^{151}Eu^{3+}$:$Y_2SiO_5$, which split into six hyperfine levels respectively in the ZEFOZ magnetic field. $|3\rangle_g$, $|4\rangle_g$, and $|3\rangle_e$ form the $\Lambda$ system for the long-lived spin-wave AFC storage. Details about the hyperfine structure and the pumping scheme can be found in the Supplementary Note 2. The AFC is prepared in the $|3\rangle_g$ level. The control pulse resonant with the $|4\rangle_g$ and $|3\rangle_e$ transfers the optical coherence into a spin wave and drives the spin wave back to the optical regime. **b** Schematic of the experimental setup. The probe and pump beam are emitted with single-mode fibers (SMF) and fiber collimators (FC) before entering the cryostat. The two beams are arranged in a non-collinear configuration. Half-wave plates ($\lambda/2$) control the polarization of the beams, and two pairs of lenses are used to obtain proper beam widths at the position of the crystal. A BS (beam splitter) with a reflection-to-transmission ratio of 8:92 is employed to efficiently collect the transmitted signal. The probe beam is reflected by the mirror at the bottom of the cryostat and coupled to a SMF for optical heterodyne detection after passing through the BS. A pair of coils placed at the two sides of the sample is driven by an arbitrary waveform generator (AWG) with the output amplified with a 300-W RF amplifier.

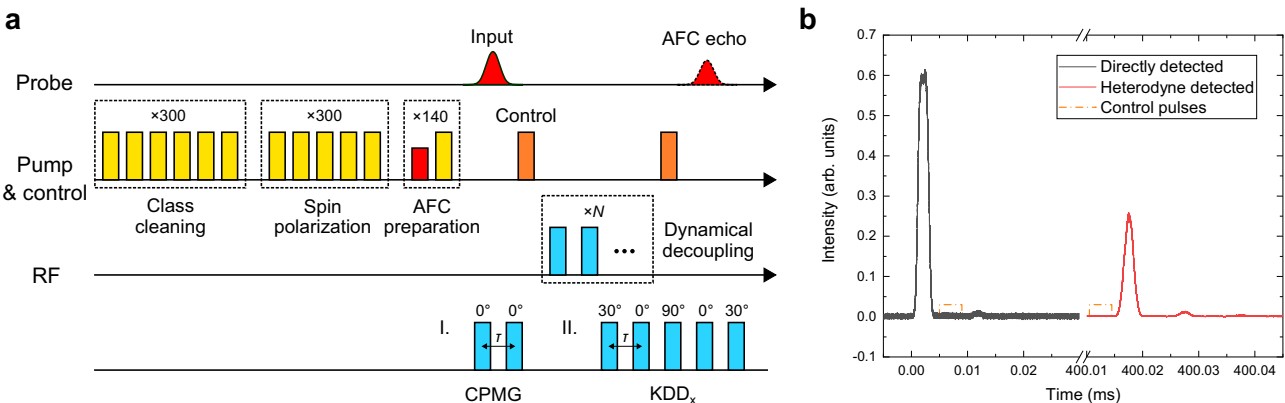

**Fig. 2 Experimental sequence and oscilloscope traces. a** The experimental sequence. The class-cleaning sequence containing six pump lights is used to pump away unwanted ions and select a single class of ions. The spin polarization sequence polarizes the population into the $|3\rangle_g$ state. Then the AFC is prepared in the $|3\rangle_g$ state while keeping the $|4\rangle_g$ empty using the $|4\rangle_g \leftrightarrow |3\rangle_e$ pump light. After the input pulse (Gaussian profile, colored with red) is absorbed, a control pulse transfers the optical excitation into a spin-wave excitation for long-term storage. Dynamical decoupling is employed for protecting the spin coherence and finally the spin excitation is transferred back to the optical regime with another control pulse. Two kinds of dynamical decoupling sequences, CPMG and KDD$_x$, are tested for spin coherence protection. **b** Oscilloscope traces of the spin-wave AFC storage with a CPMG sequence with $\tau = 100$ ms, which consists of four $\pi$ pulses. The black line shows the traces of transmission of the input mode and remaining two-level AFC echo, which are directly detected by the photodetector. The red line shows the trace of the spin-wave AFC echo after a storage of 400 ms, which is heterodyne detected. The traces are averaged four times. The dashed line indicates the position of the control pulses.

second control pulse transfers the spin coherence back to the optical excitation. The collective state will continue rephasing and emits an AFC echo when the excited-state storage time reaches $1/\Delta = 10$ μs.

Two kinds of DD sequences, i.e., CPMG and KDD$_x$, are used to protect the spin coherence in our experiments. As shown in Fig. 2a, each of them consists of sequences of $\pi$ pulses with different phase shifts, where $\tau$ is the spacing between the center of the $\pi$ pulses. As a comparison with the optical storage times, the spin coherence times with DD are also measured using RHD (see

Supplementary Note 3). CPMG and KDD$_x$ can reach a $1/e$ spin coherence lifetime of $2.68 \pm 0.06$ and $0.95 \pm 0.03$ h when $\tau = 20$ ms, respectively. For storage of light, we measure the decay of the echo with various intervals $\tau$ shown in Fig. 3a, c. The longest storage times are achieved at $\tau = 100$ ms, where CPMG and KDD$_x$ have a lifetime of $52.9 \pm 1.2$ and $33.3 \pm 1.1$ min, respectively. These results are in good agreement with the spin coherence times measured with the same $\tau = 100$ ms, which are $50.6 \pm 2.0$ and $38.2 \pm 2.0$ min (see Supplementary Note 3).

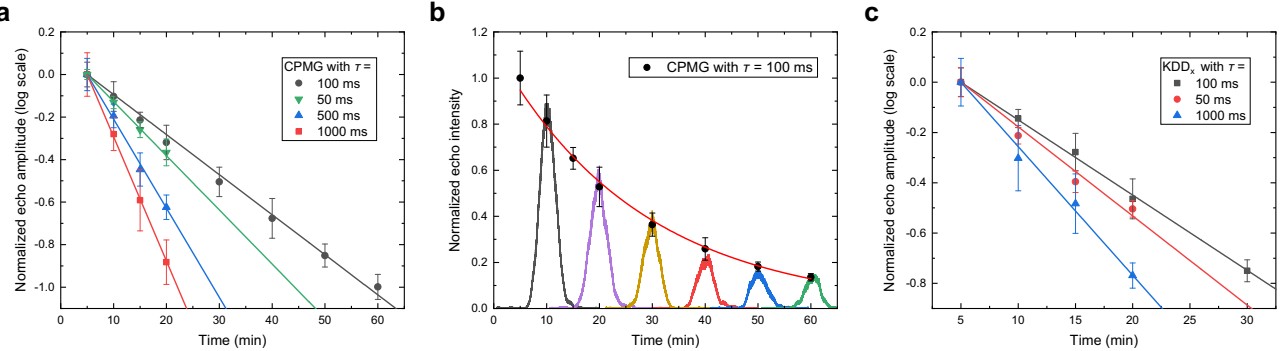

**Fig. 3 Measurements of the memory lifetime. a**, **c** Decay of the spin-wave AFC echo with the CPMG and KDD$_x$ sequences, respectively. The two DD sequences reach their longest storage lifetime of 52.9 ± 1.2 and 33.3 ± 1.1 min with a period $\tau = 100$ ms. **b** The decay of the echo intensity using CPMG with $\tau = 100$ ms versus the storage time. Some of the single-shot oscilloscope traces of the read out signals are included. Each data point is averaged four times. The error bars indicate one standard deviation of the echo area.

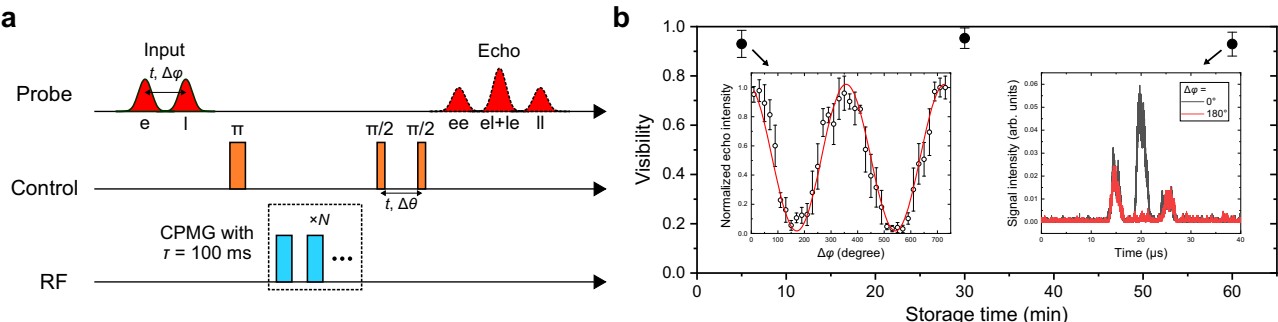

**Fig. 4 Verification of the coherent nature of the memory. a** Experimental sequence of the AFC echo interference, where e and l represent the earlier and the later, respectively. The interval of the two input pulses and that of the two read out pulses ($\pi/2$) should be the same in order to overlap the echoes (el +le) of the two inputs. **b** Visibility versus the storage time. The left inset shows an interference pattern after 5 min storage, generated by recording the intensity of the echo in the middle while varying the relative phase $\Delta\varphi$ between the two input pulses. Each data point is averaged four times, and the error bars indicate one standard deviation of the echo area. After a storage time of 5, 30, and 60 min, the coherence is well maintained with a visibility of 93.0 ± 5.5%, 95.3 ± 4.2%, and 92.9 ± 4.9%, respectively. Single-shot oscilloscope traces of the signals after the 60 min storage are presented in the right inset. Constructive and destructive interference are observed at $\Delta\varphi = 0°$ and 180°, respectively.

In Fig. 2b, we present oscilloscope traces of the spin-wave AFC storage with a CPMG sequence with $\tau = 100$ ms, which consists of four $\pi$ pulses. Traces recorded after longer storage times can be found in Fig. 3b. When $\tau$ is shorter than 100 ms, unlike the spin coherence times, the optical storage lifetimes are shortened as shown in Fig. 3a, c. This is because DD sequences with smaller intervals result in a greater heating effect of the RF coils, which broadens the optical homogeneous linewidth[28], leading to a reduction of the AFC efficiency.

The storage efficiency is analyzed as follows:

$$\eta_{total} = \eta_{AFC} \cdot (\eta_{control})^2 \cdot \eta_{spin}. \quad (1)$$

The two-level AFC efficiency $\eta_{AFC}$ without spin-wave storage and DD is 4.5%. This efficiency can be enhanced close to unity by using cavity enhancement[29] technique to achieve an efficient photon absorption. We also note that this efficiency will be reduced if a RF sequence is applied to the sample due to the heating. For example, $\eta_{AFC}$ is reduced to 2.5% after a DD sequence with $\tau = 100$ ms. The transfer efficiency of the control pulse $\eta_{control}$ is determined to be 38.5% (see Supplementary Note 2). This transfer efficiency is not very high because the polarization of the incident light cannot be parallel to D$_1$ axis (maximum absorption) of the crystal due to the special sample orientation described above. The total storage efficiencies for 5

min storage with CPMG and KDD$_x$ when $\tau = 100$ ms are 0.035% and 0.052%, respectively. $\eta_{spin}$ is the efficiency of storage in the spin states, which can be estimated from Eq. (1) to be 9.5% and 14.1% for 5 min storage with CPMG and KDD$_x$ when $\tau = 100$ ms, respectively. $\eta_{spin}$ is primarily limited by the insufficient bandwidth of the $\pi$ pulses and the inhomogeneity of the RF coils in our case. The width of the $\pi$ pulse is determined at 65.1 μs (see Supplementary Note 3). However, the inhomogeneous broadening $\Gamma_{inh}$ of the spin transition $|3\rangle_g \leftrightarrow |4\rangle_g$ of the Eu$^{3+}$ ions in the crystal is 30 kHz, which cannot be entirely covered by the bandwidth of the $\pi$ pulse $\Gamma_\pi \approx 1/(65.1 \text{ μs}) \approx 15$ kHz. The decoherence of those ions, with transition frequencies outside the bandwidth of the $\pi$ pulse, cannot be recovered by the DD sequences, thus leading to an efficiency loss. $\eta_{spin}$ is also related to the inhomogeneity of the RF field generated by the coils, because the thickness of the sample is comparable to the diameter of the coils. A more efficient spin manipulation can be achieved by RF coils with an improved homogeneity, thus increasing the storage efficiency. The total storage efficiencies can be greatly enhanced by the combination of those techniques mentioned above.

**Coherence verification**. Quantum information can be encoded into various degrees of freedom of photons, such as frequency,

time and polarization. The time-bin encoding has the particular advantage of good robustness to environmental fluctuations. Therefore, here we implement the coherent light storage with time-bin-like encoding[17]. The experimental sequence is shown in Fig. 4a. Here the AFC preparation and DD sequences are the same as that described above. In order to verify its long-term coherence protection, CPMG with an interval $\tau = 100$ ms is used to extend the spin coherence time. Two input pulses with an interval $t$ are stored in the memory and partially read out with two $\pi/2$ pulses separated by the same $t$, respectively, after the CPMG sequence. Each input pulse is read out as two AFC echoes, and therefore the later echo of the earlier input and the earlier of the later are overlapped. The relative phase $\Delta\varphi$ between the two input pulses is varied to generate the interference pattern presented in Fig. 4b while fixing the relative phase $\Delta\theta$ between the two $\pi/2$ pulses. The visibility of the interference after a storage time of 5, 30, and 60 min is $V = 93.0 \pm 5.5\%$, $95.3 \pm 4.2\%$, and $92.9 \pm 4.9\%$, corresponding to a fidelity of $F = (1 + V)/2 = 96.5 \pm 2.8\%$, $97.6 \pm 2.1\%$, and $96.4 \pm 2.5\%$, respectively, which indicates a promising application as a faithful quantum memory for time-bin qubits in the future. Compared with the longest optical storage time demonstrated so far, which is 1 min realized in $Pr^{3+}$: $Y_2SiO_5$[10], this coherent storage time is approximately enhanced by 60 times.

## Discussion

In summary, spin-wave AFC protocol is combined with ZEFOZ technique for the purpose of long-lived quantum storage of photonic qubits. An optical memory with a coherent optical storage time of 1 h and a time-bandwidth product of $3.6 \times 10^9$ is achieved using a $^{151}Eu^{3+}:Y_2SiO_5$ crystal. In order to extend this work to single-photon regime, higher efficiencies and good signal-to-noise ratios (SNR) are required. There are two sources of noise when it comes to single-photon regime. One is the coherent noise caused by the intense control pulse, which can be filtered out with spectral holes[17], Fabry–Pérot cavities[12], etc. The other is the photon noise caused by extra population in the ground state introduced by the DD sequences[30], which can be suppressed by better optical pumping and high-fidelity $\pi$ pulses. The storage time in this work can be further extended by using a stronger ZEFOZ field[23], where a higher SNR can be expected since less DD pulses are required.

## Data availability

The data that support the findings of this study are available from the corresponding author upon reasonable request.

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

## Acknowledgements

This work is supported by the National Key R&D Program of China (No. 2017YFA0304100), the National Natural Science Foundation of China (Nos. 11774331, 11774335, 11504362, 11821404, and 11654002), Anhui Initiative in Quantum Information Technologies (No. AHY020100), the Key Research Program of Frontier Sciences, CAS (No. QYZDY-SSW-SLH003), the Science Foundation of the CAS (No. ZDRW-XH-2019-1), and the Fundamental Research Funds for the Central Universities (No. WK2470000026, No. WK2470000029). Z.-Q.Z. acknowledges the support from the Youth Innovation Promotion Association CAS.

## Author contributions

Z.-Q.Z. and C.-F.L. conceived the experiment. Y.M. and Y.-Z.M. performed the experiment. Y.M. and Z.-Q.Z. wrote the paper. Z.-Q.Z., C.-F.L., and G.-C.G. supervised the project. All authors discussed the experimental procedures and results.

## Competing interests

The authors declare no competing interests.

**Additional information**

