## [Peer Review File · Nature Communications]

Reviewers' Comments:

Reviewer #1:

Remarks to the Author:

The paper "One-hour coherent optical storage in an atomic frequency comb memory" by Ma et al. describes exactly what it says in the title. The authors were able to demonstrate (and verify) coherent storage of an optical pulse of light for almost an hour in a cryogenically cooled rare-earth doped crystal using the atomic frequency comb (AFC) memory protocol. This result was a long time coming, since Zhong et al. showed in their seminal 2015 paper that the same material used in the current work, Eu:YSO, can maintain spin coherence for 6 hours under a ZEFOZ field and using dynamical decoupling. Actual optical storage in these conditions has proven elusive until now though.

While earlier this year Holzäpfel et al. ("Optical storage for 0.53 s in a solid-state atomic frequency comb memory using dynamical decoupling" NJP 2020) have demonstrated optical storage times under 1 second using the same protocol, with dynamical decoupling and in the same material (and hence should be cited in this work), combining the latter with the ZEFOZ field in order to reach hour-long storage times, is a huge achievement and deserves to be published in a high profile journal like Nature Communications.

Before I can wholeheartedly recommend this paper for publication though, I would like the authors to address a few questions/issues (in addition to including the citation above):

a) The authors write in their abstract and conclusion that "By combining this long-lived optical memory with high-speed trains (with a speed of 300 km/h), this scheme provides significantly enhanced data rate as compared to that relying on direct transmission in telecom optical fibers". While a similar tongue-in-cheek point was mentioned in the seminal Zhong paper, it is a bit irrelevant and incorrect, as the 1 MHz bandwidth of the presented memory (which has a big impact on the data rate) is much lower than the bandwidth of modern (quantum) communications (GHz). I'd argue Ma et al.'s impressive results are scientifically significant enough on their own to warrant publication in Nature Comms, even without this semi-practical justification.

b) The authors state that their doping concentration was "0.1 at.%". Does that correspond to 1000 ppm?

c) Could the authors please change the ordering of the plot legend in Fig. 3a, so that it either follows the lines on the plot or the magnitudes of tau? It would make it clearer I think.

d) The authors mention that their class-cleaning pulses and AFC structure were 1 MHz bandwidth. They then mention that the signal pulses were 2 μ s-long and the control ones 4 μ s-long. Were the signal and control pulses Fourier transform-limited or chirped to 1 MHz as well? If it's the former, then there's clearly a bandwidth mismatch between signal and control, which would lead to a decrease in the transfer efficiency η_{control} . Could the authors please clarify and comment on this?

e) On page 5 the authors write "Compared with the longest coherent storage time so far, which is one second realized in cesium vapor [Katz. et al. 2018], the coherent storage time is improved by three orders of magnitude." Considering the authors also cite older papers showing light storage for 1 minute [10,13], this comparison seems irrelevant. While it's true that [10,13] haven't explicitly shown the preservation of coherence, the protocol used in both of these (EIT) has been independently shown to be coherent, therefore [10,13], and the resultant 60-fold improvement in storage time, seem like a more fair comparison.

Reviewer #2:

Remarks to the Author:

The present manuscript reports the storage of classical light pulses using the AFC protocol in a Eu:YSO crystal. The authors employ well known dynamical decoupling techniques at a zero-first

order Zeeman point to reach storage times as long as 60 minutes. Moreover, the coherent storage of temporal modes is also demonstrated, always in the classical regime.

The present experiment is a step forward with respect to the results of the Canberra group that, back in 2015, demonstrated in the same material spin coherence times of 6 hours. While I believe that this is an important demonstration in the field of optical memories with prospects in quantum communication, I am not fully convinced that the amount of new physics and advance in understanding is such to justify the publication on Nature Communications, mainly, but not only, because all experiments are performed in the fully classical regime. Moreover, I feel that there is still room for improvement in the way the experiments are presented, as summarised in the comments listed below.

1) Considering how the present work builds up on several preparatory experiments made by other groups in the same field, I believe the bibliography is somewhat limited, especially in terms of single-photon or single-photon level storage. For example, when mentioning the record storage times achieved in previous demonstrations, I would also add the record storage times for single photon storage. Also, I would recommend at least citing the spin-wave storage of polarization qubit at the single photon level (New J. Phys. 18 013006 (2016) and the generation and multimode storage of single photons as spin waves (PhysRevLett.118.210501) demonstrated by the Geneva group in the same material used here.

2) I find the following sentence a bit confusing "Additionally, AFC memory has the advantages of wide bandwidth [19, 20] and large multimode capacity [21-24] for practical applications.". It is not clear to me whether they still speak about the spin-wave AFC, as in the previous sentence, or just the excited state AFC. I am particularly doubtful regarding the advantage of the wide bandwidth, which is true for some crystals in the excited state AFC, but not quite when the spin-wave AFC is used. First for the difficulty of having control pulses with high enough Rabi frequency and second because the hyperfine splitting of the ground state becomes a limiting factor. The citations do not help understanding because they are quite mixed.

3) Can the authors motivate the choice of the frequency shift for the heterodyne detection?

4) I think at least one full raw trace of the full spin-wave AFC protocol (input, AFC, spin wave and maybe control pulses) should be shown here, even if at shorter storage times. The only "traces" shown are some post processed segments of echo, supposedly after the demodulation of the local oscillator used for the heterodyne detection.

5) In figure 3, 4 and S3, some data points are reported with errorbars, some without. Can the authors clarify how do they obtain these data points (e.g. are they from averages or from single shot measurements?), how do they calculate the errors and why some data sets do not have errors?

6) Can the author say which is the angle between the input and the control pulse modes? With a control pulse with a power as high as 360 mW and a width of 4 μ s, I would expect the measurements to show significant leakage of the control pulse in the detection mode, while they are all surprisingly flat. Especially considering that the excited state storage time, 10 μ s, is barely sufficient to fit two control pulses. Do they apply any kind of gating/filtering?

7) The authors correctly state that Fig. 3B features high SNR. However, considering the experimental parameter, their input pulses still contain a considerable number of photons (I reckon 10^{13}). I would suggest rephrasing that part and maybe avoid using the SNR as a figure of merit, that is typically used in the single photon regime.

Throughout the manuscript there are some minor revisions to do, listed below.

- The acronym DD is never defined.

- The parameter τ is only defined in a figure. Given its importance I would define it in the main text too.

- At the end of page 3, in the sentence "For the optical storage, we measure the decay of the echo with various intervals.." I guess what the authors mean by "optical storage" is "storage of classical light", but it is a bit confusing as "optical" can be also referred to excited state storage.

- In Fig. 4b, what are the three big dots with arrows outside of the plot frames?

- In table S1, it should be stated somewhere that those are MHz.

- At the end of page 3 of the supplementary material, I guess that the transition in parenthesis is - $3/2g \leftrightarrow +3/2g$?

In the following we provide a point-to-point response to reviewers' comments. All the original comments are in black font and our reply is in blue font. The related changes in the manuscript are highlighted.

Reviewer #1 (Remarks to the Author):

The paper "One-hour coherent optical storage in an atomic frequency comb memory" by Ma et al. describes exactly what it says in the title. The authors were able to demonstrate (and verify) coherent storage of an optical pulse of light for almost an hour in a cryogenically cooled rare-earth doped crystal using the atomic frequency comb (AFC) memory protocol. This result was a long time coming, since Zhong et al. showed in their seminal 2015 paper that the same material used in the current work, Eu:YSO, can maintain spin coherence for 6 hours under a ZEFOZ field and using dynamical decoupling. Actual optical storage in these conditions has proven elusive until now though.

While earlier this year Holzäpfel et al. ("Optical storage for 0.53 s in a solid-state atomic frequency comb memory using dynamical decoupling" NJP 2020) have demonstrated optical storage times under 1 second using the same protocol, with dynamical decoupling and in the same material (and hence should be cited in this work), combining the latter with the ZEFOZ field in order to reach hour-long storage times, is a huge achievement and deserves to be published in a high profile journal like Nature Communications.

We sincerely thank the reviewer for recommendations on our manuscript and providing constructive comments to guide our revision. The NJP paper is now included as reference [28].

Before I can wholeheartedly recommend this paper for publication though, I would like the authors to address a few questions/issues (in addition to including the citation above):

a) The authors write in their abstract and conclusion that "By combining this long-lived optical memory with high-speed trains (with a speed of 300 km/h), this scheme provides significantly enhanced data rate as compared to that relying on direct transmission in telecom optical fibers". While a similar tongue-in-cheek point was mentioned in the seminal Zhong paper, it is a bit irrelevant and incorrect, as the 1 MHz bandwidth of the presented memory (which has a big impact on the data rate) is much lower than the bandwidth of modern (quantum) communications (GHz). I'd argue Ma et al.'s impressive results are scientifically significant enough on their own to warrant publication in Nature Comms, even without this semi-practical justification.

We thank the reviewer for these positive comments. We have deleted this statement according to your suggestion.

b) The authors state that their doping concentration was "0.1 at.% ". Does that correspond to 1000 ppm?

Here 0.1 at.% is equivalent to 1000 ppm. We have used the ppm notation in the main text.

c) Could the authors please change the ordering of the plot legend in Fig. 3a, so that it either follows the lines on the plot or the magnitudes of tau? It would make it clearer I think.

We thank the reviewer for pointing out this to us. We have corrected the figure according to your suggestion.

d) The authors mention that their class-cleaning pulses and AFC structure were 1 MHz bandwidth. They then mention that the signal pulses were 2us-long and the control ones 4us-long. Were the signal and control pulses Fourier transform-limited or chirped to 1 MHz as well? If it's the former, then there's clearly a bandwidth mismatch between signal and control, which would lead to a decrease in the transfer efficiency η_{control} . Could the authors please clarify and comment on this?

We are sorry that we didn't make this clear in the main text. In the Supplementary Information, we have actually pointed out that the control pulse has a hyperbolic secant profile to achieve efficient control over a bandwidth of 1 MHz. We have added this information in the lower part of page 3 in the main text.

e) On page 5 the authors write "Compared with the longest coherent storage time so far, which is one second realized in cesium vapor [Katz. et al. 2018], the coherent storage time is improved by three orders of magnitude. " Considering the authors also cite older papers showing light storage for 1 minute [10,13], this comparison seems irrelevant. While it's true that [10,13] haven't explicitly shown the preservation of coherence, the protocol used in both of these (EIT) has been independently shown to be coherent, therefore [10,13], and the resultant 60-fold improvement in storage time, seem like a more fair comparison.

We have corrected the statement according to your suggestion.

Reviewer #2 (Remarks to the Author):

We thank the reviewer for making helpful comments and reply below, in blue font, to the posed comments.

The present manuscript reports the storage of classical light pulses using the AFC protocol in a Eu:YSO crystal. The authors employ well known dynamical decoupling techniques at a zero-first order Zeeman point to reach storage times as long as 60 minutes. Moreover, the coherent storage of temporal modes is also demonstrated, always in the classical regime.

The present experiment is a step forward with respect to the results of the Canberra group that, back in 2015, demonstrated in the same material spin coherence times of 6 hours. While

I believe that this is an important demonstration in the field of optical memories with prospects in quantum communication, I am not fully convinced that the amount of new physics and advance in understanding is such to justify the publication on Nature Communications, mainly, but not only, because all experiments are performed in the fully classical regime. Moreover, I feel that there is still room for improvement in the way the experiments are presented, as summarised in the comments listed below.

The reviewer thinks that our work is an important demonstration in the field of optical memories but he/she worries about whether there are enough advances since this experiment is in the classical regime. We shall clarify that we indeed solve a number of long-standing challenges to arrive to the current result and this work represents a long-expected result in this community.

Although the optical coherence storage is in the classical regime, this work is exploratory because it is implemented in a strong ZEFOZ magnetic field, where both the ground and excited state lift their degeneracy and split into six hyperfine levels with a previously unknown structure.

Since the report of 6-hour spin coherence time by the Canberra group [Nature 517, 177 (2015)], there are growing interests in implementing long-lived memory in Eu:YSO with ZEFOZ fields. Our work is the first successful experiment and it comes after 6 years. This result was a long time coming, taking the words from the reviewer 1, because there are a number of well-known challenges before arriving to the current result.

To make this clear, we can simply compare the energy level structure reported by the Canberra group [Nature 517, 177 (2015)] and that in our work (cf. the following figure). It is clear that the excited state level structure is unknown in Nature 517, 177 (2015). In addition, the numbers and ordering in the ground states are not correct in Nature 517, 177 (2015). This is not a problem for their own work because only the single hyperfine transition at ~ 12.5 MHz is investigated at that time. However, this prevents any demonstration of an optical memory in this system. We now explicitly mentioned this point in the 3rd paragraph in the Supplementary Information (SI).

Canberra group → Our work
Nature 2015

To our knowledge, this is the first time that the complete level structure of a non-Kramers ion (such as Pr, Eu) is determined in a ZEFOZ field. This is also the first work on spin-wave AFC storage with an energy level structure that consists of 12 well-separated levels. Although the optical pumping is complicated, a Lambda system is clearly isolated with a bandwidth of 1 MHz, **which enables an unprecedented time-bandwidth product of 3.6×10^9 for an optical memory.** We now mentioned this number in the conclusion part in the main text.

The current work shares the common challenges with future single-photon works, and the methods explored in this work are valuable and can be applied to other materials in the future.

- a) Before implementing AFC storage in the ZEFOZ field, we spent lots of efforts in detecting the nuclear magnetic resonances and determining the energy level structure. We finally find that the pulsed Raman heterodyne detection is effective, even for weak hyperfine transitions. This method can provide necessary information for implementing long-lived optical storage in strong magnetic fields in rare-earth-ion doped crystals.
- b) We cut the sample with a cylindrical shape with the optical surface perpendicular to the ZEFOZ field. This method greatly reduces the experimental difficulty of alignment, which makes our work easier to be repeated by other groups. This idea can be applied to other rare-earth-ion doped crystals for easy implementation of experiments in ZEFOZ fields.

The current work is not implemented in the single-photon regime primarily because of the low expected signal-to-noise ratio. Dynamical decoupling that consists of multiple pulses introduces population noise [Journal of Modern Optics 63, 2101 (2016)], especially for long-term storage demonstrated here, which requires a large amount of pi pulses. As we already noted in our manuscript, the dynamical decoupling is not perfect because of the thick crystal used in the experiment, which will become a major limitation when the experiment comes to the single photon level. A possible solution is to employing an impedance-matched cavity to reduce the crystal size and also enhance the storage efficiency. However, this requires a significantly larger sample space inside the magnet and is beyond the scope of the current work.

In the field of long-lived optical memory, the best result demonstrated so far is 1-minute EIT storage in Pr:YSO with a ZEFOZ field, reported by the group of Thomas Halfmann [PRL 111, 033601 (2013)]. In the viewpoint published with this PRL paper (<https://physics.aps.org/articles/v6/80>), the author (Hugues de Riedmatten) wrote: "*there is clear hope that this could be a solid-state platform (Eu:YSO) able to achieve coherent light storage for tens of minutes.*" 8 years after that, our work demonstrate the coherent light storage of one hour, finally realizing this dream. **Our work enhances the optical memory time by 60 times.** AFC has been the only successful spin-wave quantum storage protocol in solid-state ensemble. **Our work improves the AFC memory time by approximately 6000 times.**

The current work is a long-expected result in the community, which provides the only solution (so far) towards the transportable quantum memory and the space-borne quantum repeater.

As a result, we believe this work meets the standard of Nature Communications and will stimulate many more following-up works in this exciting field. Meanwhile, we have carefully revised the manuscript and SI to follow the helpful comments and suggestions from the reviewer.

1) Considering how the present work builds up on several preparatory experiments made by other groups in the same field, I believe the bibliography is somewhat limited, especially in terms of single-photon or single-photon level storage. For example, when mentioning the record storage times achieved in previous demonstrations, I would also add the record storage times for single photon storage. Also, I would recommend at least citing the spin-wave storage of polarization qubit at the single photon level (New J. Phys. 18 013006 (2016)) and the generation and multimode storage of single photons as spin waves (PhysRevLett.118.210501) demonstrated by the Geneva group in the same material used here.

We have added these citations in the 4th paragraph on the first page in the main text.

2) I find the following sentence a bit confusing “Additionally, AFC memory has the advantages of wide bandwidth [19, 20] and large multimode capacity [21-24] for practical applications.”. It is not clear to me whether they still speak about the spin-wave AFC, as in the previous sentence, or just the excited state AFC. I am particularly doubtful regarding the advantage of the wide bandwidth, which is true for some crystals in the excited state AFC, but not quite when the spin-wave AFC is used. First for the difficulty of having control pulses with high enough Rabi frequency and second because the hyperfine splitting of the ground state becomes a limiting factor. The citations do not help understanding because they are quite mixed.

We have deleted the citations concerning excited-state AFC according to your suggestion.

3) Can the authors motivate the choice of the frequency shift for the heterodyne detection?

The frequency shift 43.66 MHz is chosen to make full use of the 50-MHz detection bandwidth of the photodetector. Higher frequency shifts can help reducing the $1/f$ noise.

4) I think at least one full raw trace of the full spin-wave AFC protocol (input, AFC, spin wave and maybe control pulses) should be shown here, even if at shorter storage times. The only “traces” shown are some post processed segments of echo, supposedly after the demodulation of the local oscillator used for the heterodyne detection.

We didn’t record such complete traces in our previous measurements. Therefore, we have implemented new experiments to obtain these traces and added a full raw trace of the spin-wave AFC protocol with a 4-pulse CPMG sequence in Fig. 2(b). The transmission field is directly detected, while the spin-wave echo has to be heterodyne-detected because it is too small to be directly detected by the photodiode.

The following is a trace for spin-AFC, where the black line shows a directly-detected trace after a spin-wave AFC storage without dynamical decoupling, and the red line shows a heterodyne-detected trace of the spin-AFC echo.

5) In figure 3, 4 and S3, some data points are reported with errorbars, some without. Can the authors clarify how do they obtain these data points (e.g. are they from averages or from single shot measurements?), how do they calculate the errors and why some data sets do not have errors?

We are sorry that we didn't include all the errorbars in these figures. Each data point in Fig. 3 and 4 are averaged four times, and three times in Fig. S3. We have provided this information in the caption of these figures, and added errorbars for each point. The errors are obtained by calculating the standard deviation values of the echo area.

6) Can the author say which is the angle between the input and the control pulse modes? With a control pulse with a power as high as 360 mW and a width of 4 μ s, I would expect the measurements to show significant leakage of the control pulse in the detection mode, while they are all surprisingly flat. Especially considering that the excited state storage time, 10 μ s, is barely sufficient to fit two control pulses. Do they apply any kind of gating/filtering?

In our experiment, the input and the control mode overlaps at the sample position with an angle of ~ 30 mrad. We have added this information in the 2nd paragraph on page 2 in the revised main text.

There is no significant leakage of the control pulses when a silicon photodiode is used as the detector (new Fig. 2(b)). If there is no DD applied, the second control pulse is overlapped with the two-level AFC echo, but not the spin-wave AFC echo, in the time domain, since the spin-wave AFC echo appears at 4 μ s later after the two-level AFC echo.

There are two reasons that can explain why the detected traces are so flat:

- a) Due to the specially designed cryostat used in our experiment, the crystal is placed deep inside the cryostat. The total optical path of the input field is as long as 4 meters, which leads to a good separation between the spatial modes of the input and control field. The only filter used in our experiment is a single-mode fiber that collects the double-transmitted input field.
- b) There is no leakage observed in the heterodyne detected traces, because the demodulator used for heterodyne detection is set at 43.66 MHz with a bandwidth of 1 MHz, and can only demodulate signals that are close to the input field in frequency. However, the frequency of the control field has a difference of 12.46 MHz with respect to the input field, which makes it invisible in the detected traces.

7) The authors correctly state that Fig. 3B features high SNR. However, considering the experimental parameter, their input pulses still contain a considerable number of photons (I reckon 10^{13} ?). I would suggest rephrasing that part and maybe avoid using the SNR as a figure of merit, that is typically used in the single photon regime.

We decided to remove this statement since the “high SNR” is obvious in the figure.

Throughout the manuscript there are some minor revisions to do, listed below.

- The acronym DD is never defined.

Corrected.

- The parameter τ is only defined in a figure. Given its importance I would define it in the main text too.

Corrected.

- At the end of page 3, in the sentence “For the optical storage, we measure the decay of the echo with various intervals..” I guess what the authors mean by “optical storage” is “storage of classical light”, but it is a bit confusing as “optical” can be also referred to excited state storage.

Corrected.

- In Fig. 4b, what are the three big dots with arrows outside of the plot frames?

We are sorry that we did not make this clear in the caption. The three dots denote the visibility versus the storage time. We have revised the caption of this figure.

- In table S1, it should be stated somewhere that those are MHz.

The unit “MHz” is stated in the caption.

- At the end of page 3 of the supplementary material, I guess that the transition in parenthesis is $-3/2g \leftrightarrow +3/2g$?

Corrected.

We thank the referee for reading our manuscript so carefully. We have carefully checked and improved the presentations to make sure that the results are easily accessible for a wide audience.

Reviewers' Comments:

Reviewer #1:

Remarks to the Author:

The authors have successfully addressed all of my comments/concerns, therefore I am happy to recommend the paper for publication in Nature Comms.

Reviewer #2:

Remarks to the Author:

All my comments and concerns have been satisfactorily addressed by the authors in the revision.

REVIEWERS' COMMENTS

Reviewer #1 (Remarks to the Author):

The authors have successfully addressed all of my comments/concerns, therefore I am happy to recommend the paper for publication in Nature Comms.

We sincerely thank the Reviewer for the effort and expertise that you contribute to reviewing.

Reviewer #2 (Remarks to the Author):

All my comments and concerns have been satisfactorily addressed by the authors in the revision.

We sincerely thank the Reviewer for the effort and expertise that you contribute to reviewing.